# Novel Cerium(IV) Coordination Compounds of Monensin and Salinomycin

**DOI:** 10.3390/molecules28124676

**Published:** 2023-06-09

**Authors:** Nikolay Petkov, Ivayla Pantcheva, Anela Ivanova, Radostina Stoyanova, Rositsa Kukeva, Radostina Alexandrova, Abedullkader Abudalleh, Petar Dorkov

**Affiliations:** 1Faculty of Chemistry and Pharmacy, Sofia University St. Kliment Ohridski, 1164 Sofia, Bulgaria; aivanova@chem.uni-sofia.bg; 2Institute of General and Inorganic Chemistry, Bulgarian Academy of Sciences, 1113 Sofia, Bulgaria; radstoy@svr.igic.bas.bg (R.S.); rositsakukeva@yahoo.com (R.K.); 3Institute of Experimental Morphology, Pathology and Anthropology with Museum, Bulgarian Academy of Sciences, 1113 Sofia, Bulgaria; rialexandrova@hotmail.com (R.A.); alkader78mah@yahoo.com (A.A.); 4Research and Development Department, Biovet Ltd., 4550 Peshtera, Bulgaria; p_dorkov@abv.bg

**Keywords:** polyether ionophore, monensin, salinomycin, Ce(IV) mixed-ligand complexes, computational modeling, antibacterial activity, cytotoxic effect

## Abstract

The largely uncharted complexation chemistry of the veterinary polyether ionophores, monensic and salinomycinic acids (HL) with metal ions of type M^4+^ and the known antiproliferative potential of antibiotics has provoked our interest in exploring the coordination processes between MonH/SalH and ions of Ce^4+^. (1) Methods: Novel monensinate and salinomycinate cerium(IV)-based complexes were synthesized and structurally characterized by elemental analysis, a plethora of physicochemical methods, density functional theory, molecular dynamics, and biological assays. (2) Results: The formation of coordination species of a general composition [CeL_2_(OH)_2_] and [CeL(NO_3_)_2_(OH)], depending on reaction conditions, was proven both experimentally and theoretically. The metal(IV) complexes [CeL(NO_3_)_2_(OH)] possess promising cytotoxic activity against the human tumor uterine cervix (HeLa) cell line, being highly selective (non-tumor embryo Lep-3 vs. HeLa) compared to cisplatin, oxaliplatin, and epirubicin.

## 1. Introduction

Nowadays, the increasing number of resistant bacterial species and the insufficient activity in the field of antibiotic development tend to prevail over the numerous attempts of humanity to keep itself healthy and away from bacterial infections [1]. The strict measures which are enforced over clinical trials and the heavy approval procedures make the way of a new antibacterial drug from the flask to the shelf very long and expensive. One helpful approach in changing the activity of already-known compounds towards better performance is their modification using the tools and the advantages of coordination chemistry [2,3,4,5,6,7,8]. The inclusion of bioactive molecules in metal complex species may alter their biopharmaceutical properties, leading to tremendously increased drug efficacy.

A notably dynamic area is research oriented to the development of anticancer therapeutics. The requirements that one compound should meet to prove its efficacy and safety are numerous, an important one being the selective drug cytotoxicity against tumor cell lines. The morphological changes in the neoplastic cells, such as receptor over-expression, change in membrane fluidity, and pH ranges in the surrounding interstitial area make this task possible but not easy to solve. The increasing expectations of modern medicine to achieve better cancer treatment oblige scientists to work intensively on drug targeting and new molecule development. Coordination chemistry could again offer various approaches to achieve the goal set. Furthermore, the metal complexes have been on the chemotherapeutic protocols for a long time and their efficacy has been demonstrated [9,10,11,12,13].

In the last few decades, some representatives of the polyether ionophores group have become a subject of increasing scientific interest. They are natural antibiotics that undergo pseudo-cyclization since the carboxylic group takes part in intramolecular hydrogen bond formation with the hydroxyl group(s) at the opposite end of the molecule. Thus, the formed cavity is highly hydrophilic and can host an alkaline cation or a water molecule. In addition to their main application—the treatment of parasitic disease coccidiosis—the polyether ionophores are also thought to possess anticancer activity and antibacterial effect [14,15,16,17,18,19,20,21,22,23,24]. These findings make them attractive compounds when searching for new applications of already-known therapeutics. Some of the attempts at improving the activity of this antibiotic class are focused on functionalizing the carboxylic group, in some cases followed by further complex formation [25,26,27,28,29]. An alternative approach is to use the native ionophores as ligands directly to obtain various metal complexes. Both pathways have proven to be possible and reliable tools for enhancing the biological activity of parent compounds.

The two most studied polyether ionophores—monensin and salinomycin (Figure 1)—are the target antibiotics in the present study. Several case-study reports present salinomycin as a promising chemotherapeutic agent in human medicine with noteworthy success in cancer treatment [15,30,31,32,33]. It targets not only the tumor cells but also the cancer stem cells which are responsible for metastasis processes. On the other hand, salinomycin is an attractive chelating agent and there are several studies dedicated to the synthesis of different metal-based derivatives [34,35,36,37]. Monensin is also a subject of extensive studies although it has not been used in human medicine yet. An overview of the published results on the activity of monensin and its complexes shows that they exhibit pronounced in vitro activity over some Gram-positive bacteria and human/animal tumor cell lines, incl. drug-resistant cells [38,39,40,41,42,43,44,45,46,47].

The studies on the biological activity of cerium(IV) reveal that CeO_2_ is a subject of interest in nano-scaled science due to its ability to form nano-sized particles which demonstrate potential to be applied in many scientific fields [48,49,50,51,52]. The tetravalent cerium compounds are widespread in organic synthesis and catalysis due to the strong oxidizing potential of the Ce(IV)/Ce(III) couple in acidic aqueous media. On the contrary, basic conditions and anionic oxygen-bearing ligands, as well as low dielectric solvents, are able to stabilize Ce(IV) due to strong coordination of oxophilic cerium ion by O-donor ligands [53]. In 2017 So and Leung summarized the advances in the coordination chemistry of Ce(IV) complexes supported by various mono-, di- and polydentate ligands [54]. Some of the already-known coordination species of Ce(IV) exhibit pronounced cytotoxicity comparable to or higher than that observed for cisplatin and oxaliplatin. Thus, the complex of 8-hydroxyquinolone derivative [CeL_4_] is effective against human ovarian (SK-OV-3), liver (BEL-744), and lung (NCl-H460) cancer cells, with good selectivity compared to a normal liver cell line (HL-7702) [55]. The Ce(IV) complex of 3-MBTMPD imine derivative [CeL(NO_3_)_2_(H_2_O)_2_] possesses significant cytotoxicity in cervical (HeLa) and breast (MCF-7) cancer cells [56], while the Ce(IV)-bearing dicarboxylate [CeL_3_]^2−^ shows activity against breast cancer (MCF-7), colon adenocarcinoma (HT-29) and lymphocyte (HT-60) cells [57].

To combine the properties of cerium ion in its highest oxidation state (+4) on the one hand, and those of the two members of polyether ionophores on the other, we studied the complexation process between polyether ionophores and Ce(IV) at different reaction conditions. Our interest in these systems was, in addition, provoked by the lack of studies on the coordination chemistry of monensin and salinomycin with metal ions of type M(IV). The isolated new complexes were characterized by a set of spectral and thermal methods; most of the structures were modeled by quantum-chemical calculations. The effect of Ce(IV) ion on the antibacterial and cytotoxic activity of both ionophorous antibiotics was evaluated against Gram-positive aerobic bacterial strains and cultured human tumor/non-tumor cell lines.

## 2. Results and Discussion

In the present paper, we report the synthesis and the structural characterization of novel Ce(IV) complexes of the polyether ionophores, monensic acid A (monohydrate) and salinomycinic acid, bearing, in addition, hydroxide (**1**) or hydroxide and nitrate (**2**) ligands. The experimental data revealed the formation of two types of mononuclear coordination species whose composition and structure depend primarily on the metal-to-ligand molar ratio used in the synthetic protocols. Reactions proceed in an acetonitrile solution at high excess of weak bases which ensures the antibiotic deprotonation and the neutralization of the acidic cerium(IV) salt ((NH_4_)_2_[Ce(NO_3_)_6_]) but also serves as a hydroxide ions source.

At metal-to-ligand molar ratio of 1:2, the complexes [CeL_2_(OH)_2_] (**1a**, L = Mon^−^; **1b**, L = Sal^−^) form, while at equimolar conditions (M(IV):L = 1:1) and at the excess of NH_4_NO_3_, the species of composition [CeL(NO_3_)_2_(OH)] (**2a**, L = Mon^−^; **2b**, L = Sal^−^) were isolated. The spectral properties of the new coordination compounds are studied in solid state (IR, EPR, TG–DTA, TG–MS, elemental analysis) and in solution (IR, NMR, UV-Vis). The structure of complex **1a** is optimized with density functional theory (DFT), while that of **2a**–**b** is modeled by a molecular dynamic approach followed by quantum-chemical calculations. The computed spectral features of **1a** and **2a**–**b** are compared to the experimentally observed. As shown below, some of the novel Ce(IV) complexes possess promising anticancer activity compared to the parent antibiotics.

### 2.1. Characterization of Complexes ***1a**–**b***

The IR spectrum of monensic acid (MonH × H_2_O) in KBr pellets (Table 1, Appendix A) contains several intense bands in the range of 4000–3000 cm^−1^, assigned to the stretching vibrations of the water molecule placed in the hydrophilic antibiotic cavity, and of both hydroxyl groups at the ligand tail engaged in the formation of intramolecular H-bonds. The two bands observed in the range of 1750–1600 cm^−1^ are due to the stretching carbonyl vibration of the COOH group and the deformation vibration of water. Moving from solid state to solution (CHCl_3_), some shift and minor broadening of the signals attributed to ν_OH_ (4000–3000 cm^−1^) are detected, but, in general, the overall spectra are preserved in both phases. The observed differences can be explained by the conformational changes of the antibiotic molecule in solution, which affect the hydrogen bond strength, but also allow the formation of new ones.

In the IR spectrum of **1a** (solid state) (Table 1, Appendix A), the multiple bands of the ligand in the 4000–3000 cm^−1^ range are replaced by a single broad band. The shape of the signal differs significantly from that observed for already-known [M(Mon)_2_(H_2_O)_2_] complexes [38,39,40], assuming the absence of water molecules in the composition of **1a**. The intense ν_C=O_ band of the parent ligand disappears in the spectrum of **1a** at the expense of two new intense signals which arise from the asymmetric and symmetric stretching vibrations of the carboxylate function. The difference between the two vibrations (Δ$\tilde{\nu }$ = 138 cm^−1^) points to the monodentate coordination mode of the COO^−^-moiety and to the possible formation of a macrocyclic structure, as in the case of divalent mononuclear bis-monensinates [58]. The IR spectrum of **1a** in CHCl_3_ is identical to that recorded in the solid state, except for the negligible protonation of monensinate anion due to the impurities of HCl dissolved in the solvent.

The IR spectrum of salinomycinic acid (SalH, Table 2, Appendix A) in the solid state is characterized by a broad band in the range of 4000–3000 cm^−1^ due to the stretching vibrations of the hydroxyl groups in the molecule. The single intense signal centered at 1710 cm^−1^ is attributed to the ν_C=O_ arising from the carboxylic and the isolated ketone functions of the ligand, while the band of low intensity at ca. 1640 cm^−1^ is assigned to the stretching C=C vibrations. Like monensic acid, the solution IR spectrum of SalH in CHCl_3_ is practically the same as observed in the solid state.

In the IR spectrum of **1b** (KBr) (Table 2, Appendix A), the broad band of ν_OH_ in SalH does not significantly change, which is why no essential structural information regarding the OH-containing functional groups can be retrieved. On the other hand, the intensity of the band at 1710 cm^−1^ significantly diminishes and is accompanied by the appearance of two new bands due to the deprotonation of SalH during the complexation process. From the close value of Δ$\tilde{\nu }$ = 141 cm^−1^ as calculated for the complex **1a**, it can be concluded, that the Ce(IV) cation is placed in a similar coordination environment in both **1a** and **1b** species. The solution IR studies in CHCl_3_ reveal that the structure of **1b** is preserved both in solid and solution states despite the minor degree of protonation of the salinomycinate monoanion.

The complexes **1a**–**b** absorb in the UV-Vis range (MeOH solutions), where an intense asymmetric band in the range of 200–500 nm is observed (Appendix A). The electronic configuration of Ce(IV) resembles that of Xe^0^, so such an absorbance (a_270_ = 1.71 L·g^−1^·cm^−1^, **1a**; a_282_ = 2.65 L·g^−1^·cm^−1^, **1b**) can only be explained by O → Ce charge transfers occurring in the novel coordination compounds. This fact confirms that the internal coordination shell of **1a**–**b** in solution primarily comprises O-donor atoms.

The participation of both antibiotics in **1a**–**b** is confirmed as well by corresponding NMR studies in CDCl_3_ solution. The ^1^H-NMR spectra of **1a**–**b** are significantly broadened compared to the acidic antibiotics, corroborating the inclusion of the heavy metal ion into their structure. Considerable differences between the ^1^H-NMR spectra of MonH × H_2_O and complex **1a** are detected (Appendix A). Thus, the narrow and intense peak at 6.25 ppm in the spectrum of monensic acid (assigned to the OH-groups) shifts to a lower field (6.76 ppm in **1a**). Simultaneously, the signal attributed to CH-2 (2.62 ppm, MonH × H_2_O) significantly broadens down to the baseline in **1a**, confirming the proximity of the heavy metal to the carboxylate function of the antibiotic. Based on the recorded NMR changes, one can conclude that the monensinate anion serves as a bidentate ligand in the formation of **1a** via the carboxylate group located at the head of the antibiotic molecule and the hydroxyl group placed at its opposite (tail) end. In contrast, the proton spectrum of SalH is very complicated due to the overlapping signals of some characteristic protons placed at the two opposite ends of the organic molecule, and for that reason, no substantial body of information can be retrieved comparing the spectra of SalH and **1b** (Appendix A). Although the ^1^H-NMR spectra of SalH and **1b** are more intricate, the similar coordination behavior of both antibiotics in their known coordination compounds can be transmitted to their identical binding in Ce(IV) complexes **1a**–**b**.

The X–band solid state EPR spectra of complexes **1a**–**b** are presented in Figure 2a,c. In the spectra of both samples, narrow signals (∆H_pp_ ≈ 1.2 mT) of low intensity were observed with g-values in the range of 2.004–2.030. The signal with g ≈ 2.011 overlaps with the signal at g = 2.004 and becomes visible only at low temperatures. The measurements at 100 K showed higher signal intensity, according to the Currie–Weiss law. The EPR parameters of the discussed signals are attributed to defect centers in the studied complexes. The signal with g = 2.004 could relate to the presence of an oxygen-centered radical [59], while the signals with g ≈ 2.011 and g ≈ 2.030 are most likely components of an asymmetric signal characteristic for radicals, containing the pair Ce^4+^-oxygen [60]. It should be noted that, according to the EPR analysis, the concentration of defect centers in samples **1a**–**b** is rather low and would not affect their overall behavior.

The experimental data for **1a**–**b** presented above lead to the conclusion that monensin and salinomycin deprotonate in the presence of a weak base and bind in a bidentate manner to the cerium(IV) ions via two functional groups, forming a “classical” macrocyclic structure through the “head” monodentate carboxylate and the “tail” hydroxyl functions. Such a coordination mode is typical for these two ionophores; in addition, most of their known complexes also contain water ligands, hosted by the antibiotics in their hydrophilic cavity. To check the presence of coordinated water, further thermal studies on the target samples **1a**–**b** under argon atmosphere were performed—TG–DTA (weight decrease, accompanied by the corresponding endo-/exothermic effects, Figure 3 and Figure 4, up) and TG–MS (H_2_O/CO_2_/NO loss, Figure 3 and Figure 4, bottom).

Three endothermic peaks below 250 °C are detected in the TG–DTA curve of monensic acid (Figure 3a). The first of them corresponds to the melting point of MonH (102 °C). The second is due to the release of water molecule bound in the ligand cavity (165 °C). The third endothermic peak relates to the beginning of ligand decomposition (220 °C) confirmed by the simultaneous loss of H_2_O (*m/z* 18) and CO_2_ (*m/z* 44) in the TG–MS curve (Figure 3b). In the TG–DTA curve of complex **1a** (Figure 3c), no intense endothermic peaks below 200 °C are registered which points to the absence of coordinated water molecule(s). The endothermic peak at ca. 200 °C correlates with the melting point of the complex (196 °C), followed by its further gradual decomposition (TG–MS, Figure 3d).

The TG–DTA curve of SalH (Figure 4a) shows some significant differences compared to MonH × H_2_O, despite the similar structure of the two antibiotics. The absence of intense endothermic peaks below 200 °C indicates that salinomycinic acid does not carry coordinated water. The endothermic peak at 213 °C accounts for carbon dioxide and water release (TG–MS, Figure 4b) due to the ligand decomposition. In the TG–DTA curve of complex **1b** (Figure 4c), intense endothermic peaks below 200 °C are also not observed. The weak endothermic peak at 128–150 °C agrees with the melting point of the complex (142–144 °C), followed by its decomposition (Figure 4d) at a lower temperature compared to species **1a**.

The loss of water below 100 °C is detected for all the compounds, although they were dried and kept in a desiccator before the analysis. These water molecules are weakly bound and do not relate either to the structural characteristics of the antibiotics or the coordination environment of cerium(IV) ions in complexes **1a**–**b**.

In summary, we detected the presence of cavity-hosted water molecule in monensic acid monohydrate but could not confirm its inclusion in the closed “head-to-tail” salinomycinic acid structure. In addition, all thermogravimetric results exclude the presence of water ligands in the composition of complexes **1a**–**b**. The thermal studies performed confirm that the novel **1a**–**b** coordination species do not contain water molecules in their primary coordination shell.

TEM analysis of **1a**-**b** was performed using ethanol solutions of complexes, deposited on a carbon support. The coordination compounds appear to be amorphous samples with particle size of 200–300 nm (Figure 5a,b). An even distribution of cerium, carbon, and oxygen on the sample surface is observed (Figure 5c–e), proving the homogeneity of the isolated complexes in the solid phase.

Based on stoichiometry and spectral properties of **1a**–**b**, their structures were assumed to resemble those of monensin and salinomycin complexes with divalent metal ions, i.e., two deprotonated antibiotics bind in a bidentate mode to the metal(IV) center. To achieve the overall electroneutrality of these complex species, we also imply the participation of two hydroxide ions originating from the weak base excess used during preparation procedures. The initial structure of complex **1a** for the density functional theory (DFT) calculations was constructed using the crystallographic data for [Ni(Mon)_2_(H_2_O)_2_] [40] by substituting the metal ion for Ce(IV) and removing one hydrogen from each water molecule bound in the monensin cavity. The geometry of the thus-arranged complex species [Ce(Mon)_2_(OH)_2_] was optimized according to the procedures described in Section 3.4 (Figure 6). The metal–oxygen distances vary from 2.04 Å in Ce-OH^−^ to 2.35 and 2.45 Å in COO^−^-Ce and Ce-OH, respectively. The *cis*-oriented organic ligands occupy the equatorial plane of the complex and the hydroxide ions placed at axial positions complete the internal coordination shell. The geometry can be described as a distorted octahedron with the following bond angles: 77.2° (COO^—^-Ce-OH), 87.4° (OH-Ce-OH), 118.2° (COO^−^-Ce-COO^−^), and 178.4° (OH^−^-Ce-OH^−^). The vibrational analysis (Table 1) of the scaled frequencies follows the trends observed in the IR spectrum of complex **1a**. Due to the very limited crystal structure data for salinomycin and its derivatives, we refrained from optimizing the possible construct of complex **1b**, but the closeness in the spectral behavior of **1a**–**b** suggests their isostructurality. Considering the composition [CeL_2_(OH)_2_] (L = Mon, **1a**; L = Sal, **1b**), the calculated molar extinction coefficients in MeOH (ε_270_ = 2590 L·mol^−1^·cm^−1^, **1a**; ε_282_ = 4434 L·mol^−1^·cm^−1^, **1b**) agree with the proposed charge transfer transitions occurring from O-donor atoms to the heavy metal ion.

### 2.2. Characterization of Complexes ***2a**–**b***

The complexes **2a**–**b** possess similar spectral behavior to that already observed for species **1a**–**b**. The main differences between the two types of coordination species arise from the composition of **2**. The elemental analysis indicates that one heavy metal ion is bound to one deprotonated antibiotic, two nitrates, and one hydroxide ion. The main spectral features of **2a**–**b** are summarized below:

(i) The IR spectra of solids **2a**–**b** (Table 1 and Table 2, Appendix A) confirm the deprotonation of monensic and salinomycinic acids, respectively, but disclose the bidentate coordination mode of the carboxylate function due to the smaller values of Δ$\tilde{\nu }$ for the corresponding asymmetric and symmetric C=O vibrations: Δ$\tilde{\nu }$ = 91 cm^−1^, **2a**; Δ$\tilde{\nu }$ = 101 cm^−1^, **2b**. In addition, the spectra contain new bands in the range 1560–1280 cm^−1^, attributed to the stretching vibrations of nitrate anions. The difference between the two signals (Δ$\tilde{\nu }$ = 216 cm^−1^, **2a**; Δ$\tilde{\nu }$ = 224 cm^−1^, **2b**) indicates the participation of NO_3_^−^ ions as ligands directly bound to the Ce(IV) center. The band at 378 cm^−1^ (**2a**) and 380 cm^−1^ (**2b**), respectively, is due to the deformation vibration of nitrate anions, engaged in the formation of four-membered chelate structures, thus assuming their bidentate coordination mode. The overall IR spectra of both complexes in CHCl_3_ are comparable to those in the solid state. As species **1a**–**b**, a negligible protonation degree of the antibiotics is also observed.

(ii) The UV-Vis spectra of **2a**–**b** in MeOH (Appendix A) contain an asymmetric band arising from O → Ce charge transfer transitions (a_275_ = 2.37 L·g^−1^·cm^−1^, **2a**; a_300_ = 3.07 L·g^−1^·cm^−1^, **2b**).

(iii) Related to complex **1a**, the ^1^H-NMR signal of the hydroxyl proton attached to C-25 in **2a** is broadened, retaining its position. The same tendency in the CH-2 signal of monensic acid A is also observed in **2a** (Appendix A). As in the case of **1b**, the ^1^H-NMR spectrum of **2b** cannot be used to derive significant structural information on salinomycin binding functions (Appendix A).

(iv) The EPR spectrum of complex **2a** (Figure 2b) fully corresponds to these of samples **1a**–**b** (g = 2.004, 2.011, 2.030). For comparison, in the spectrum of complex **2b** (Figure 2d), the signal with g = 2.004 is missing but two other signals are registered with effective g-factors 2.002 and 1.960, respectively. While the first signal is attributed to a carbon-centered radical [59], the second one is informative of the presence of Ce^3+^ ions in the studied sample. Characterized by fast spin lattice relaxation, normally Ce^3+^ ions remain EPR-silent at temperatures over 10–20 K. Nevertheless, the radical comprising the pair Ce^3+^-O results in an appearance of an anisotropic signal with two components (g_⊥_ ≈ 1.96 and g_II_ ≈ 1.93) [60], which is easily detected at higher temperature. In the spectrum of sample **2b**, only the perpendicular part of this signal is clearly visible. In line with complexes **1a**–**b**, the intensity of all observed signals for **2a**–**b** is quite low.

(v) The TG–DTA and TG–MS studies of **2a** (Figure 3e,f) and **2b** (Figure 4e,f) exclude the presence of water ligands in the internal coordination shell of the complexes. In contrast to **1a**–**b**, the TG–DTA curves of **2a**–**b** comprise an exothermic peak at 258 °C (**2a**) and 153 °C (**2b**), respectively, due to the oxidation processes with the participation of nitrate anions. The data reveal that the degradation of complexes begins before their melting with an intense release of H_2_O, CO_2_, and NO (TG–MS). Compared to species **2a**, these processes occur at lower temperature in **2b**, but at a slower rate, and the active decomposition of **2b** takes place over 350 °C.

(vi) The TEM analysis confirms the amorphous character of **2a**–**b** and the even distribution of elements on their surface (Figure 7).

Combining the spectral properties of **2a**–**b** with elemental analysis data, we suggest that cerium(IV) cation is bound to three types of ligands—ionophore, nitrate and hydroxide anions. Their potential multiple binding variances do not allow the construction of an initial structure to be further optimized by the DFT approach. For that reason, we applied molecule dynamics accompanied by subsequent quantum chemical calculations (for details, refer to Section 3.4). A total of eight (monensin) and three (salinomycin) conformers were sufficiently populated at 298 K (0.1–39.3%, **2a**; 19.7–47.7%, **2b**) after cluster analysis and DFT optimization. These structures were used for vibrational spectra calculations and the determination of donor atoms set that comprises the primary coordination shell of **2a**–**b**.

The results show that complexes **2a**–**b** represent mononuclear cerium(IV) coordination compounds with a metal ion placed nearby the hydrophilic antibiotic cavity (Figure 8). They are the first example of metal-containing ionophores whose structure is close to that of alkaline complexes of monensin and salinomycin. A single monensinate (**2a**) or salinomycinate (**2b**) is bound to the cerium(IV) cation in a polydentate coordination manner, and the electroneutrality is ensured by one hydroxide and two nitrates. The coordination number of Ce(IV) ion is nine in **2a**–**b**, but it is realized in a different way depending on the length of the polyether ionophore chain. Thus, monensin serves as a pentadentate ligand at the expense of mono- and bidentate nitrates (**2a**), forming a folded macrocyclic structure with the participation of the “head-to-tail” located carboxylate and hydroxyl functions. In contrast, salinomycin is a tetradentate ligand, but the “tail” hydroxylic group does not coordinate to Ce(IV) in **2b** and the two nitrates are bound bidentately. In both complexes, the hydroxide anion is attached in a monodentate mode. The quantum chemical calculations reveal the following complex composition: [Ce(Mon)(η_2_-NO_3_)(NO_3_)(OH)] for **2a**, and [Ce(Sal)(η_2_-NO_3_)_2_(OH)] for **2b**, respectively.

Selected bond lengths observed in **2a**–**b** are presented in Table 3, using the conformers with the highest population (39.3 and 25.9% for **2a**; 47.7 and 32.7% for **2b**). All distances are significantly smaller than the sum of cerium(IV) cation (1.85 Å) and oxygen anion (1.52 Å) van der Waals radii and in such a way can be discussed as reliable values. The Boltzmann distribution at 298 K was calculated over all conformationally and energetically unequal structures. The energies of the most significant transitions in the vibrational spectra of **2a**–**b** were averaged using the weights obtained from the population of the different structures and these values corroborate the experimental data well (Table 1 and Table 2). The calculated molar coefficients from the UV-Vis spectra are ε_275_ = 2253 L·mol^−1^·cm^−1^ for **2a**, and ε_300_ = 3165 L·mol^−1^·cm^−1^ in the case of **2b**.

### 2.3. Biological Activity of Polyether Ionophores and Their Ce(IV) Coordination Species ***1**–**2***

To evaluate the effect of Ce(IV) cation on biological performance of monensin and salinomycin, we assessed the antimicrobial activity of ionophores and complexes **1**–**2** against a set of three Gram-positive bacteria, namely *B. subtilis*, *B. cereus* and *K. rhizophila*. Representatives of Gram-negative microorganisms were not included in the present study since their cell walls do not permit the penetration of molecules with molecular weights over 600 and thus, these microorganisms are not susceptible to the action of ionophores and their metal complexes [61].

The antibacterial assay performed demonstrates that the four new Ce(IV) coordination compounds exhibit variable activity against the tested bacterial strains compared to the parent acidic antibiotics. The starting cerium(IV) salt takes no effect at 1 mg/mL, while the ionophores and their metal derivatives are effective within a wide range of concentrations varying from 3.91 µg/mL to 500 µg/mL (Table 4). Considering the composition of the tested substances, we recalculated their minimum inhibitory concentration in terms of µM, which precisely accounts for the chemical structure of the targets.

Among the studied microorganisms and in the framework of the particularly applied protocol, the representative of the *Kocuria* genus possesses the lowest sensitivity upon the treatment with target compounds, followed by the strains of *B. subtilis* and *B. cereus*. Monensic acid appears to be less toxic than SalH against *K. rhizophila*, but the order is reversed in the case of *B. subtilis* and *B. cereus*. The inclusion of Ce(IV) cation into the structures of **1a**–**b** and **2a**–**b** leads to different results depending on the bacterial strain, with structurally similar complexes causing reduced toxicity in some cases or, conversely, increased activity compared to the starting antibiotics. Based on MIC data, the following hierarchical orders of decreasing antibacterial activity can be constructed:

*K. rhizophila*:   **1b** > **2b** > SalH > MonH × H_2_O > **1a** > **2a**

*B. subtilis*:     **1a** > **1b** > MonH × H_2_O > **2b** > SalH > **2a**

*B. cereus*:       **1b** ≈ **1a** ≈ MonH × H_2_O > **2b** > SalH > **2a**.

The data show that the nitrate-containing monensinate complex **2a** is the least toxic among the studied samples, while the corresponding salinomycinate counterpart **2b** possesses strong to moderate activity (Table 4). The complex species **1a**–**b** also display diverse effects on visible bacterial growth, and no general conclusions for the metal ion influence on polyether ionophores’ antibacterial efficacy can be derived based on the limited data set obtained in this study.

The reason for paired analogous compounds (MonH/SalH; **1a**/**1b**; **2a**/**2b**) to display different activity against Gram-positive microorganisms is a matter of much more in-depth research that goes beyond the tasks of the present study. At present, these experimental data illustrate that the properties of a given drug candidate cannot be unequivocally “automatically translated” from one biological object to another, and purposeful research is required for each individual system being studied. In this respect, the question of how the new **2a**–**b** complexes of Ce(IV) (and with a different structure compared to the known so far monensinates and salinomycinates) will affect the development of non-pathogenic/pathogenic bacterial strains, tumor, and non-tumor cell lines remains open. Furthermore, the acute toxicity on in vivo model should be evaluated in the future as an essential part of such versatile research, which could shed light on the possibilities for new applications of polyether ionophores (and the corresponding metal derivatives) outside of their antiparasitic properties, approved in veterinary medicine.

To answer some questions, as stated above, the novel complex species **2a** and **2b** were included in short- (MTT, 24–72 h; double staining with AO/PI, 72 h) and long-term (colony-forming method, 26 days) experiments to assess their potential against human tumor HeLa (uterine cervix) compared to the acidic ionophores. The effect of MonH × H_2_O, SalH and **2a**–**b** on cell viability of human non-tumor embryo cell line Lep-3 was also evaluated. The tested compounds decrease viability and/or proliferation of HeLa and Lep-3 in a concentration- and time-dependent manner, which is highly pronounced after 72 h treatment (Appendix A).

The CC_50_ values of the compounds of interest, derived from the corresponding dose–response curves (72 h exposure), and the calculated selectivity indices (SI) are summarized in Table 5. For comparative purposes, the data on the potential of conventional chemotherapeutics cisplatin, oxaliplatin, and epirubicin are also included [62]. The results (in µM) reveal the following hierarchy, starting from the most effective compound:

HeLa:  **2a** > MonH × H_2_O > SalH > **2b** > oxaliplatin > cisplatin > epirubicin;

Lep-3:  epirubicin > cisplatin > oxaliplatin > Mon × H_2_O > **2a** > SalH, **2b**.

The selectivity index of both antibiotics and cerium(IV) complexes **2a**–**b** highlights their therapeutic efficacy as potential anticancer agents with selectivity superior to that of the traditionally used drugs (Table 5). The most favorable compound is the complex, [Ce(Mon)(η_2_-NO_3_)(NO_3_)(OH)] (**2a**). The present results reveal the potential of monensin, salinomycin, and their metal modifications as promising molecules for target therapy of malignant diseases.

The results obtained are exciting not only because the tested metal complexes express more pronounced cytotoxic activity in cancer HeLa cells than the commercially available antitumor drugs and are less toxic for non-tumor Lep-3 cells. Salinomicyn has been recognized as an antitumor stem cell agent, which has greatly attracted the interest of the scientific community and medical oncologists. The reason is that cancer stem cells, which represent a small part of the tumor population, play an important role in tumor initiation and progression, including metastasis. Moreover, cancer stem cells are characterized by high resistance to radiation and chemotherapy, thus making it difficult for the therapy to succeed [63]. Further studies are needed to determine the potential antitumor activity of the metal complexes we tested, their mechanism of action, and their ability to attach cancer stem cells as well as the best route of administration.

To further investigate the properties of MonH × H_2_O, SalH and **2a**–**b**, we performed follow-up studies on cytopathological changes of HeLa cells induced by the tested compounds after 72 h treatment (Figure 9). The test was completed using double staining with acridine orange (AO) and propidium iodide (PI). The untreated HeLa (control) appears as a dense monolayer, and the intact cells possess bright green nuclear and weak cytoplasmic fluorescence (Figure 9a). The treatment of the uterine cervix with target compounds (applied in 0.5 µg/mL: MonH × H_2_O, SalH, **2a**–**b**, and 1 µg/mL: Mon × H_2_O) results in a highly reduced monolayer compared to the control, bright green nuclear fluorescence, swollen cells, vacuolization, and chromatin condensation (Figure 9b,c,e,g,h). Besides the abovementioned effects, the additional increase of ionophore concentration (5 µg/mL Mon × H_2_O; 1 µg/mL SalH) leads also to the detection of a number of apoptotic cells (Figure 9d,f).

The double staining with AO and PI corroborates the MTT assay, thus proving the cytotoxic activity of the tested compounds and their ability to stimulate apoptosis. The detailed elucidation of the mechanism of cell death induced by metal complexes needs further in-depth research. However, the obtained preliminary data in this regard are important, because for reaching an effective antitumor therapy, it is necessary to aim and to achieve apoptosis–the cell’s natural mechanism for death [64,65].

The ability of MonH × H_2_O, SalH and **2a**–**b** to suppress the 3D colony-formation of HeLa cells is expressed as the effective colony inhibitory concentration (CIC, μM)–the lowest concentration at which the compounds completely inhibit the tumor cell growth. The experimental data rank the tested samples in order of their decreasing toxicity as follows: **2b** (4.85 µM) > **2a** (5.26 µM) > MonH × H_2_O (7.25 µM) > SalH (13.31 µM).

The rapid removal of cancer cells from the body (through surgery or a “killing” therapeutic strategy) is extremely important because it minimizes the possibility of cancer cell spread, disease progression, and selection of therapy-resistant tumor cell clones. In laboratory conditions, the information obtained about the rapid antitumor effect of the tested potential antineoplastic agents comes from conducting short-term experiments (24–72 h). The purpose of the long-term experiments we carried out was to verify and confirm the cytotoxic efficiency of the target metal complexes and the irreversibility of their effect over time. In addition, the use of 3D cell cultures provides more adequate information about the influence of substances on the biological behavior of cancer cells, and to a higher degree, predicts the results of their application in vivo [66,67]. The data from short-term experiments with monolayer 2D-cultures and the long-term tests with 3D cell colonies were found to be positively correlated which illustrates the potential of cytotoxic/antitumor properties of the complexes studied.

The biological assays performed reveal that MonH and SalH, as well as their mixed-ligand cerium(IV) complexes [Ce(Mon)(η_2_-NO_3_)(NO_3_)(OH)] (**2a**) and [Ce(Sal)(η_2_-NO_3_)_2_(OH)] (**2b**) can be discussed as prospective anticancer agents. The novel cerium(IV) coordination species **2a** were found to be highly selective between non-tumoral Lep-3 and tumoral HeLa cell lines. Both **2a** and **2b** show better performance inhibiting the HeLa 3D-colony formation at lower concentration levels in contrast to the non-coordinated parent ligands. Thus, monensinate and salinomycinate, bearing a metal ion in a higher oxidation state (+4), appear to be promising metal-based therapeutics in the treatment of oncogenic diseases.

## 3. Experimental

### 3.1. Materials

Sodium complexes of monensin (MonNa) and salinomycin (SalNa) were kindly provided by Huvefarma Ltd. (Peshtera, Bulgaria). The acidic forms of the antibiotics (MonH × H_2_O and SalH) were prepared according to the literature protocol [68]. The reagents (NH_4_)_2_[Ce(NO_3_)_6_], Et_4_NOH (40% in H_2_O), Et_3_N, NH_4_OH, NH_4_NO_3_, conc. HNO_3_, acetonitrile, methanol, and anhydrous chloroform of p.a. grade were purchased from local suppliers and CDCl_3_–from Deutero Gmbh (Kastellaun, Germany).

Dulbecco’s modified Eagle’s medium (D-MEM) and fetal bovine serum were purchased from Gibco-Invitrogen (Oxford, UK). Dimethyl sulfoxide (DMSO), propidium iodide, acridine orange, and trypsin were obtained from AppliChem GmbH (Darmstadt, Germany); purified agar (Difco) and thiazolyl blue tetrazolium bromide (MTT) were from Sigma-Aldrich Chemie GmbH (Schnelldorf, Germany). All other chemicals of the highest purity commercially available were purchased from local agents and distributors. All sterile plastic ware was from Orange Scientific (Braine-l’Alleud, Belgium).

### 3.2. Physico-Chemical Methods

Infrared (IR) spectra were recorded on Nicolet 6700 FT-IR spectrometer, Thermo Scientific (Madison, WI, USA). The samples were analyzed as KBr (4000–400 cm^−1^) and CsI (600–250 cm^−1^) pellets or as solutions in CHCl_3_ (4000–850 cm^−1^). The electronic (UV-Vis) transitions were evaluated using Shimadzu UV-1800 (Kyoto, Japan) in the range of 200–1000 nm. The electron paramagnetic resonance (EPR) analyses were performed on Bruker BioSpin EMXplus10/12 EPR spectrometer (Karlsruhe, Germany) working at 9.4 GHz. The nuclear magnetic resonance (NMR) spectra were obtained using Bruker Avance III HD 500 MHz instrument (Karlsruhe, Germany). Thermogravimetry studies (TG-DTA/MS) were conducted on Setaram Labsys Evo 1600 (25–600 °C) with a heating rate of 10 K/min in argon flow (Caluire-et-Cuire, France). The apparatus is equipped with an Omnistar GSD 301 O_2_ mass spectrometer, Pfeiffer Vacuum. The transmission electron microscopy (TEM) images were observed using JEOL JEM-2100 with EDS detection (Oxford Instruments X-max 80T, Oxfird, UK). The C, H, O, N analysis was performed on an organic elemental analyzer vario MACRO cube (Elementar analysensysteme GmbH, Stuttgart, Germany). The metal content was determined on Perkin-Elmer SCIEX-ELAN DRC-e ICP-MS (Watham, MA, USA) after wet digestion of samples with conc. HNO_3_ and using appropriate standards.

### 3.3. Synthesis of Complexes ***1**–**2***

**1a**: An acetonitrile solution (5 mL) containing 0.2 mmol MonH × H_2_O (137.8 mg) and 0.2 mmol Et_4_NOH (40% in H_2_O; 76 μL) was mixed with 0.1 mmol (NH_4_)_2_[Ce(NO_3_)_6_] (54.8 mg in 5 mL acetonitrile). To the resulting yellow opalescent solution, an excess of NH_3_ was added, and the mixture was stirred for 15 min. Its precipitation in water affords the isolation of a pale-yellow solid phase. Yield: 108.9 mg (72%). Composition [Ce(Mon)_2_(OH)_2_] (C_72_H_124_O_24_Ce), MW 1513.88 g/mol. Calc.: H%, 8.26; C%, 57.12; Ce%, 9.26. Found: H%, 6.47; C%, 54.07; Ce%, 9.34. m. p. 192–198 °C.

**1b**: An acetonitrile solution (5 mL) of 0.2 mmol SalH (150.2 mg) and 0.8 mmol Et_3_N (112 µL) was mixed with 0.1 mmol (NH_4_)_2_[Ce(NO_3_)_6_] (54.8 mg in 5 mL acetonitrile) and was stirred for 30 min. The precipitation of the reaction mixture in water affords the isolation of a pale-yellow solid phase. Yield: 113.6 mg (68%). Composition [Ce(Sal)_2_(OH)_2_] (C_84_H_140_O_24_Ce), MW 1674.14 g/mol. Calc.: H%, 8.43; C%, 60.27; Ce%, 8.37. Found: H%, 7.53; C%, 57.52; Ce%, 7.42. m. p. 142–144 °C.

**2a**: The complex was obtained by mixing 0.2 mmol MonH × H_2_O (137.8 mg), 0.2 mmol Et_4_NOH (40% in H_2_O; 76 μL) and 0.2 mmol (NH_4_)_2_[Ce(NO_3_)_6_] (109.6 mg) in acetonitrile/water (1:1). The addition of 0.4 mmol NH_4_NO_3_ (32.0 mg) and NH_3_ excess, followed by precipitation in water leads to the isolation of yellow precipitates. Yield: 146.5 mg (77%). Composition [Ce(Mon)(NO_3_)_2_(OH)] (C_36_H_63_N_2_O_18_Ce), MW 951.00 g/mol. Calc.: H%, 6.57; C%, 45.47; N%, 2.95; Ce%, 14.73. Found: H%, 5.22; C%, 49.05; N%, 3.31; Ce%, 14.16.

**2b**: The addition of 0.2 mmol (NH_4_)_2_[Ce(NO_3_)_6_] (109.6 mg, in 5 mL acetonitrile) to 5 mL acetonitrile solution containing 0.2 mmol SalH (150.2 mg) and 0.8 mmol Et_3_N (112 µL) was followed by the subsequent addition of 0.2 mmol NH_4_NO_3_ (32.0 mg). The complex was precipitated with water to produce a yellow solid phase. Yield: 150.8 mg (73%). Composition [Ce(Sal)(NO_3_)_2_(OH)] (C_42_H_70_N_2_O_18_Ce), MW 1031.13 g/mol. Calc.: H%, 6.84; C%, 48.92; N%, 2.72; Ce%, 13.59. Found: H%, 5.60; C%, 49.79; N%, 2.25; Ce%, 13.19. m. p. 162–164 °C.

### 3.4. Computational Studies

The geometry optimization of complex **1a** was performed in vacuo with the B3LYP functional [69,70,71,72] using Grimme D3 correction [73] and 6–31G basis set. For the cerium ion, the Stuttgart RSC 1997 pseudopotential was used [74,75]. Vibrational analysis was done to prove that a minimum was reached. The vibrational frequencies were scaled by a factor of 0.958 [76].

A combined molecular dynamics/quantum chemical approach was used to study the structure of the newly synthesized mixed-ligand complexes with a stoichiometry of metal ion and ionophore 1:1 (**2a**–**b**) in chloroform solutions. Initial separation of monensinate and salinomycinate anions into four parts was done to properly calculate their atomic charges for the molecular dynamics simulations. Each fragment was subjected to a conformational search at 298 K in vacuo [77]. The resulting set of conformations was further processed with cluster analysis employing the Jarvis–Patrick method with a cut-off distance in the range of 0.135–0.17 nm to extract the conformationally different structures [78]. The atomic charges of the whole structures were obtained using the RESP procedure, averaging the full set of conformations obtained after the cluster analysis [79,80,81]. As starting structures for both ligands, the crystallographic data for their sodium complexes were used [82,83]. The coordinates and charges of monesinate and salinomycinate anions are presented in Appendix A.

The model systems of **2a**–**b** were constructed based on their composition derived from the experimental data. The lack of parameters for cerium(IV) ion, but the closeness of its ionic radius to that of thorium(IV), allowed us to substitute Ce^4+^ for Th^4+^, for which data are available [84]. The size of the box (with a length of the cubic box edges of 3.3 nm) was calculated for a 50 mM solution and periodicity in the three dimensions of space was applied throughout the molecular dynamics simulations. Due to the insolubility of the complexes in water, the simulations were carried out in explicitly modeled chloroform [85,86]. The OPLS-AA force field [87] was used for atomistic molecular dynamic simulations. Berendsen barostat and thermostat were employed to keep constant pressure (1 bar) and temperature (298 K) [88]. Standard molecular dynamics algorithms were applied [77]. The initial energy minimization and heating stages were followed by 10 ns of relaxation and 100 ns of the production stage. The structures from the last stage, collected at every 0.1 ns, were subjected to cluster analysis as described above with 0.04 nm cut-off for complex **2a** and 0.14 nm for complex **2b**, respectively. The resulting representative conformations were optimized as described for the structure of **1a**. The vibrational spectra were calculated to prove that a minimum was reached for all structures. The vibrational frequencies were scaled by the same factor of 0.958.

The conformational search was performed with the Hyperchem 7.0 program package [89]. The cluster analysis and the molecular dynamics simulations were done using Gromacs 5.1.2 [90], the charge distribution was computed with Amber 8 [91], and the quantum chemical calculations were processed with Gaussian 16 [92].

### 3.5. Antibacterial Test

The antibacterial effect of both ionophores, complexes **1**–**2** and (NH_4_)_2_[Ce(NO_3_)_6_] was assessed using the double layer agar hole diffusion method [93]. Three Gram-positive aerobic microorganisms were used as test strains (*B. subtilis*, *B. cereus* and *K. rhizophila*). They were provided by the National Bank for Industrial Microorganisms and Cell Cultures (NBIMCC, Bulgaria): *B. subtilis*–strain ATCC 6633/NCIMB 8054/NBIMCC1709; *B. cereus*–strain ATCC 11778/NCIMB 8012/NBIMCC 1085; *K. rhizophila*–strain ATCC 9341/NBRC 12708/DC2201.

The activity of compounds was evaluated as their minimum inhibitory concentration (MIC, µM)—the lowest concentration which inhibits visible bacterial growth. The target samples were prepared as stock solutions in DMSO at 1 mg/mL and were doubly diluted down to a concentration of 61 ng/mL; DMSO served as a negative control.

Nutrient agar (NA; 5 g/L peptone, 1.5 g/L HM peptone B (beef extract), 1.5 g/L yeast extract, 5 g/L NaCl, 15 g/L agar, final pH 7.4) was dissolved in water (2.8 g/100 mL) and sterilized at 120 °C (1 atm, 20 min). A preculture of each microorganism was grown in NA (30 °C, 24 h, aerobic conditions). Several morphologically similar colonies were suspended in sterile water until the turbidity of the inoculum at 650 nm reached that of McFarland 4 standard (1.2 × 10^9^ cfu/mL). The inoculum (1.5%) was then suspended in NA; the sterile (10 mL) and the inoculated (10 mL) agar were consecutively placed in Petri dishes. The holes (d = 6 mm) punched after agar solidification were filled with the tested solutions (20 µL). The diameter of inhibition zones was measured on the 24th hour after incubation at 30 °C. All determinations were performed in triplicate and confirmed by three separate experiments. All equipment and culture media were sterile.

### 3.6. Cytotoxicity Assays

In the current investigation, non-tumor human embryos (Lep-3) and human uterine cervix (HeLa) permanent cell lines were used as model systems. The cell lines were acquired from the National Center of Infectious and Parasitic Diseases Laboratory of Cell Cultures in Sofia, Bulgaria. In D-MEM media supplemented with 5–10% fetal bovine serum, 100 U/mL penicillin, and 100 µg/mL streptomycin, the cells were cultivated in monolayer cultures. The cultures were maintained at 37 °C in an incubator with humidified CO_2_. The adherent cells for routine passages have been separated using a solution containing 0.05% trypsin and 0.02% EDTA. The exploratory cell development phase has been used for the research.

#### 3.6.1. MTT Test

The cells were seeded in 96-well flat-bottomed microplates at a density of 1 × 10^4^ cells per well for 24 h, or until subconfluent (between 60 and 70%). Phosphate buffered saline (PBS, pH 7.2) was used to wash the cells from monolayers before covering them with a medium that contained various concentrations of the chemicals under test (0.05, 0.1, 0.5, 1, 5, and 10 µg/mL). Four to six wells were filled with each solution. As controls, cells that had been grown on an unaltered medium were used. The thiazolyl blue tetrazolium bromide (MTT) test was used to determine the compounds’ effect on cell viability and proliferation after 24, 48, and 72 h of incubation.

The MTT colorimetric assay of cells survival was performed according to Mossman procedure [94]: extraction with a solution of absolute ethanol and DMSO (1:1, vol/vol) to dissolve the blue MTT formazan after three hours of incubation with MTT solution (5 mg MTT in 10 mL D-MEM) at 37 °C under 5% CO_2_. At 540 nm, the optical density was determined using an automated microplate reader (TECAN, SunriseTM, Grödig, Austria). For each concentration, the relative cell viability has been computed as a percent of the untreated control (100% viability). The effective cytotoxic concentration of the substances CC_50_ (µg/mL, µM) that causes a 50% reduction in cell viability was calculated from these concentration-response curves using Origin 6.1. An average of three independent assays is represented by each data point.

#### 3.6.2. Double Staining with Acridine Orange (AO) and Propidium Iodide (PI)

Using AO and PI double staining, the capacity of the tested compounds to cause cytopathological changes has been studied [95]. On 6-well plates, the cells were cultured on coverslips with the target substances (0.5, 1, 5 µg/mL) present. Cells that were not treated were used as controls. The coverslips were taken out and cleaned with PBS for two minutes after 72 h of incubation. The cells were given equal amounts of fluorescent dyes containing AO (10 µg/mL in PBS) and PI (10 µg/mL in bidestilled water).

Within 30 min of the fluorescence color fading, the freshly stained cells have been placed on a glass slide and examined under a fluorescence microscope (Leica DM 500B, Wetzlar, Germany). The double staining has been used to determine the degree of apoptosis, the formation of bright green nuclei with chromatin condensation (dense green regions), and/or orange nuclei with chromatin condensation suggesting early or late apoptotic cell death, respectively.

#### 3.6.3. Colony Forming Method

24-well microplates were used to layer tumor cells (10^3^ cells/well) suspended in 0.45% pure agar in a D-MEM medium containing various concentrations of the compounds (0.5, 1, 5, 10, 20 µg/mL). Using an inverted microscope (Carl Zeiss, Oberkochen, Germany), the presence or absence of colonies was detected over 26 days. The concentration at which the investigated compounds totally inhibit the tumor cells from growing in semi-solid media is known as the colony inhibitory concentration (CIC, μM).

#### 3.6.4. Statistical Analysis

Three duplicates of each experiment were conducted. The mean and standard error of the mean has been used to present the data. One-way analysis of variance (ANOVA) was used to compare the statistical differences between the control and treatment groups, and then the Dunnett post hoc test was performed.

## 4. Conclusions

Novel coordination compounds of the veterinary polyether ionophorous antibiotics, monensic acid A (MonH) and salinomycinic acid (SalH) are isolated under non-aqueous conditions. The experimental data reveal that the reaction of (NH_4_)_2_[Ce(NO_3_)_6_] with the antibiotics at a 1:2 molar ratio leads to the formation of isostructural complexes of composition [CeL_2_(OH)_2_] (**1a**, L = Mon^−^; **1b**, L = Sal^−^), whereas the metal-to-ligand molar ratio of 1:1 shifts the process towards the isolation of [CeL(NO_3_)_2_(OH)] (**2a**, L = Mon^−^; **2b**, L = Sal^−^). The complexation reactions proceed in acetonitrile solutions in the presence of a weak base excess; in the case of species **2a**–**b**, the addition of an extra amount of nitrate anions is also required. The two polyether ionophore monoanions bind to cerium(IV) cations in a bidentate (**1a**–**b**) or polydentate coordination mode (**2a**–**b**). The weak base excess compensates the acidity of the cerium(IV) salt (NH_4_)_2_[Ce(NO_3_)_6_] used in the preparation procedure of **1**–**2**, but apparently also serves as a source of hydroxide ions, which bind to the metal(IV) center within their primary coordination shell. In contrast to complex **1**, the coordination species **2** contain in addition mono- and bidentate nitrate anions (**2a**) or two bidentate nitrates (**2b**).

The antibacterial tests showed different sensitivities of the target Gram-positive strains to the action of MonH, SalH and the corresponding metal modifications. Compounds which are similar in composition and structure display non-identical activity against the same microorganism. The reasons for the observed diverse effect require more intensive research to discern, which is not the main goal of the present work.

The cytotoxicity assays of both polyether ionophores and the novel complexes **2a**–**b** towards human uterine cervix (Hela) and non-tumor human embryonic cell line (Lep-3) reveal that complexes [CeL(NO_3_)_2_(OH)] are much more effective and selective (SI > 3) compared to the approved chemotherapeutics, cisplatin, oxaliplatin, and epirubicin.

Thus, the combinatorial approach, linking the coordination chemistry of physiologically active molecules at both the experimental and theoretical level with the evaluation of biological properties of the corresponding complex compounds, is an effective tool when looking for more powerful antibacterial or anticancer chemotherapeutics based on the repurposing of known “old” drugs.

## Figures and Tables

**Figure 1 molecules-28-04676-f001:**
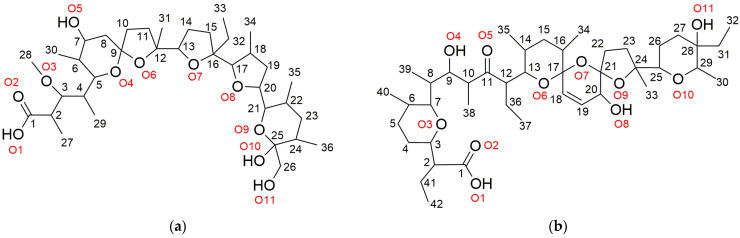
Chemical structure and numbering scheme of carbon (black) and oxygen (red) atoms in (**a**) monensic acid (MonH) and (**b**) salinomycinic acid (SalH).

**Figure 2 molecules-28-04676-f002:**
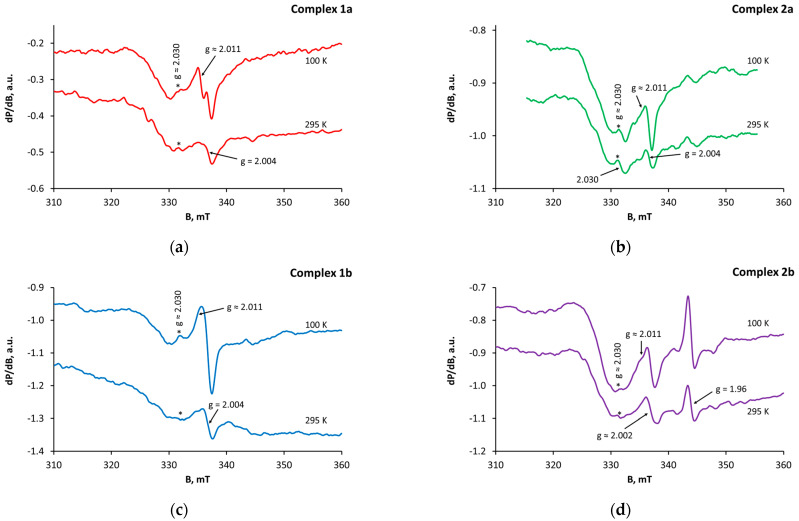
X–band solid-state EPR spectra of complexes (**a**) **1a**; (**b**) **2a**; (**c**) **1b**; (**d**) **2b** at 295 and 100 K.

**Figure 3 molecules-28-04676-f003:**
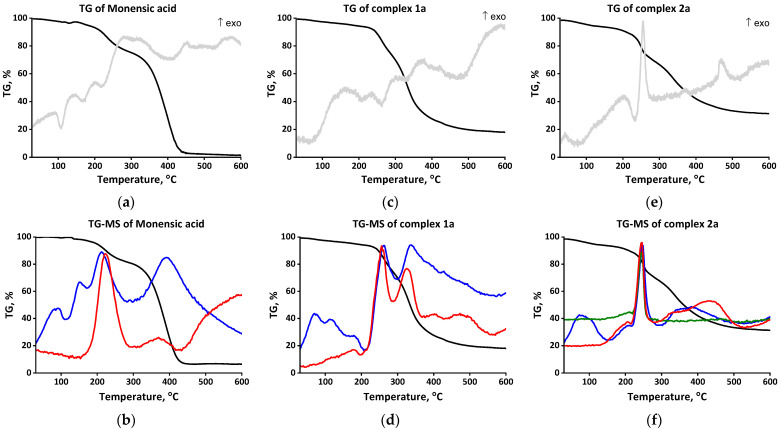
TG–DTA (up) and TG–MS (bottom) curves of MonH × H_2_O (**a**,**b**) and complexes **1a** (**c**,**d**)/**2a** (**e**,**f**). Color code: weight decrease (black); endo- and exothermic effects (grey); H_2_O loss (blue); CO_2_ loss (red), NO loss (olive).

**Figure 4 molecules-28-04676-f004:**
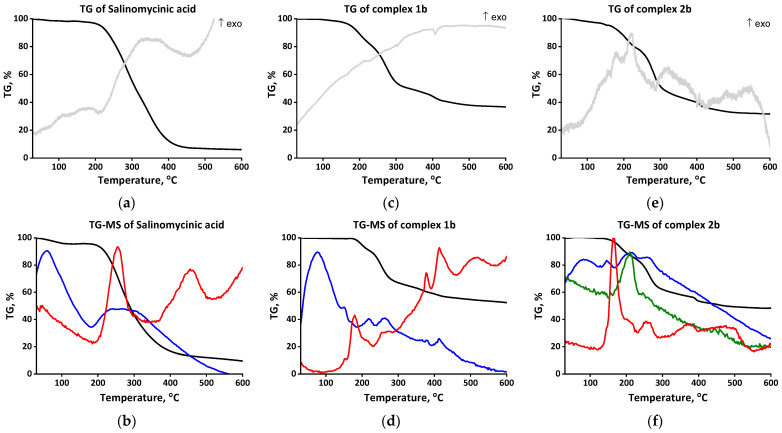
TG–DTA (up) and TG–MS (bottom) curves of SalH (**a**,**b**) and complexes **1b** (**c**,**d**)/**2b** (**e**,**f**). Color code: weight decrease (black); endo- and exothermic effects (grey); H_2_O loss (blue); CO_2_ loss (red), NO loss (olive).

**Figure 5 molecules-28-04676-f005:**
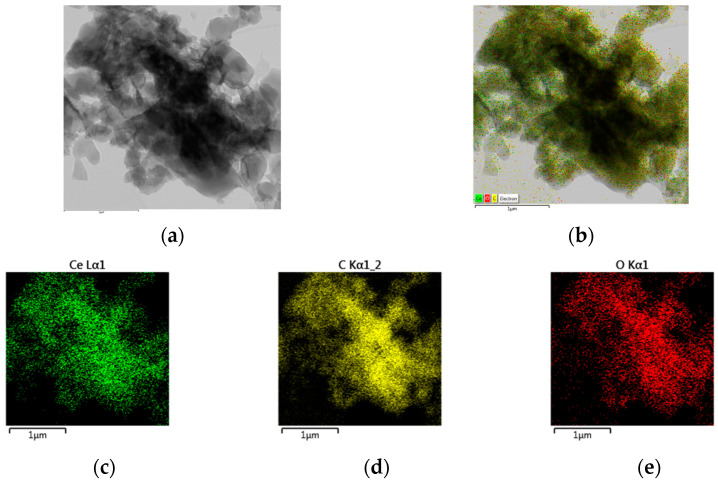
TEM-images (**a**,**b**) and element distribution (**c**–**e**) of complex **1b** taken as representative.

**Figure 6 molecules-28-04676-f006:**
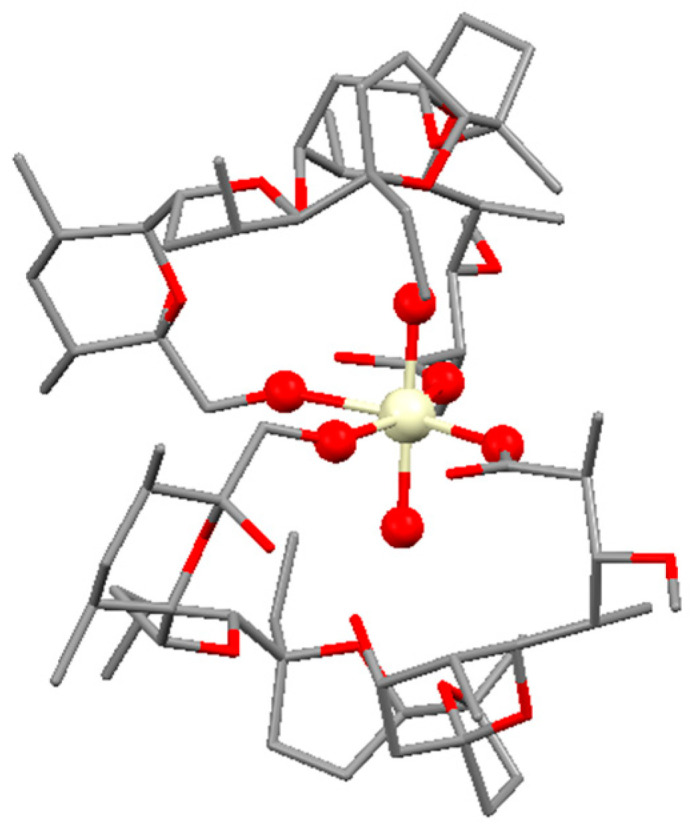
Optimized structure of the complex [Ce(Mon)_2_(OH)_2_], **1a** (hydrogens are omitted for clarity). Color code: C—grey, O—red, Ce—light yellow.

**Figure 7 molecules-28-04676-f007:**
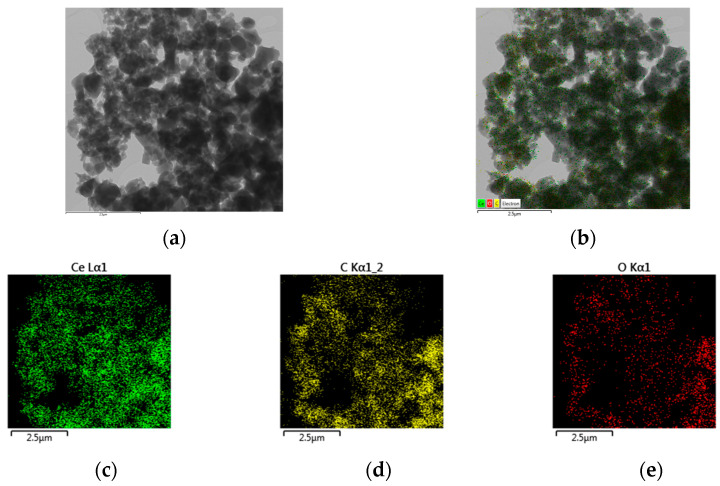
TEM-images (**a**,**b**) and element distribution (**c**–**e**) of complex **2a** taken as representative.

**Figure 8 molecules-28-04676-f008:**
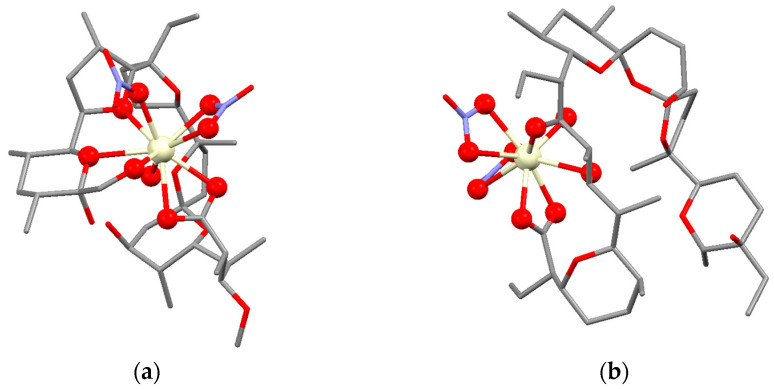
Optimized structures of complexes **2a** (**a**) and **2b** (**b**). Hydrogens are omitted for clarity. Color code: C—grey, O—red, N—violet, Ce—light yellow.

**Figure 9 molecules-28-04676-f009:**
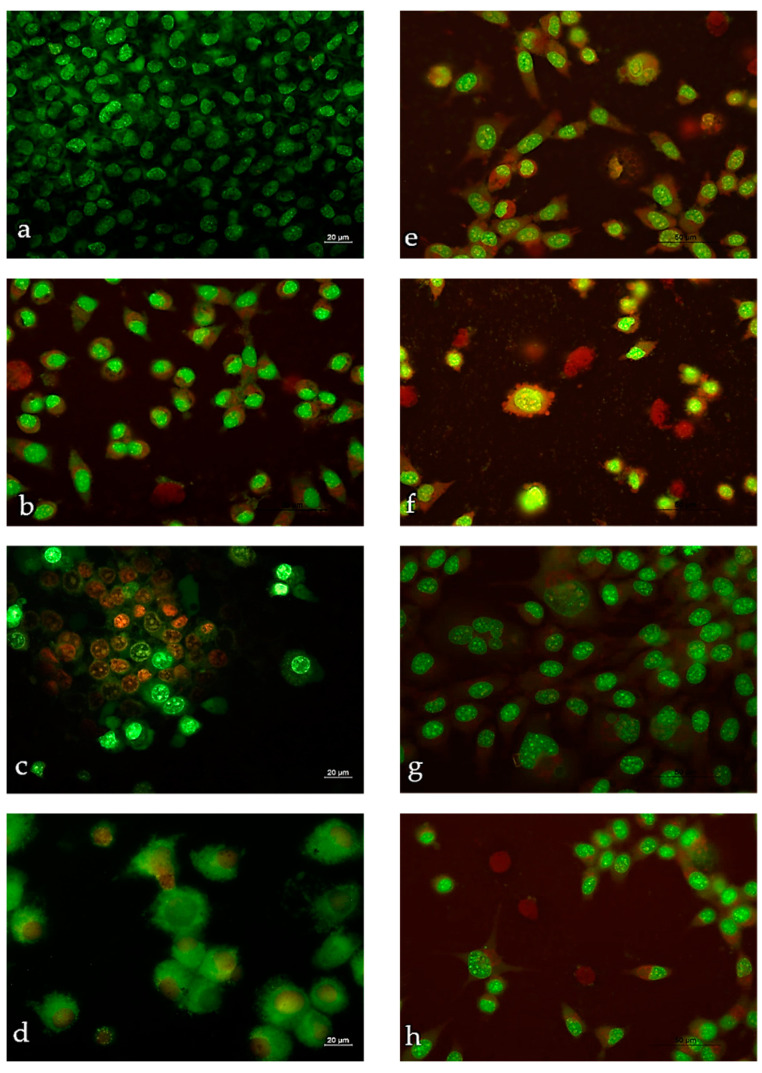
Cytopathological changes in HeLa cells after double staining with AO/PI (72 h): (**a**) untreated cells; (**b**) MonH × H_2_O—0.5 mg/mL; (**c**) MonH × H_2_O—1 mg/mL; (**d**) MonH × H_2_O—5 mg/mL; (**e**) SalH—0.5 mg/mL; (**f**) SalH—1 mg/mL; (**g**) **2a**—0.5 mg/mL; (**h**) **2b**—0.5 mg/mL (bar = 20 µm).

**Table 1 molecules-28-04676-t001:** Important IR bands (cm^−1^) of monensic acid and complexes **1a**/**2a** in KBr pellets and CHCl_3_ solution (the theoretically predicted scaled values are given in parentheses).

Vibration	MonH × H_2_O		1a	2a
	KBr	CHCl_3_	KBr/CHCl_3_	KBr/CHCl_3_
$\nu_{OH}^{asym}$_,_ H_2_O	3531	3518	-	-
$\nu_{OH}^{sym}$_,_ H_2_O	3461	3467	-	-
$\nu_{OH}$_,_ OH	3336	3293	3454	3454
$\nu_{C=O}$, COOH	1709	1707	-	-
$\nu_{C=O}^{asym}$, COO^−^	-	-	1552 (1571, 1564)	1505 (1498)
$\nu_{C=O}^{sym}$, COO^−^	-	-	1414 (1385)	1414 (1419)
δ_HOH_, H_2_O	1627	1621	-	-
$\nu_{NO}$, NO_3_^−^	-	-	-	1555 (1586, 1521)
$\nu_{NO}$, NO_3_^−^	-	-	-	1289 (1282, 1222)
ν_MO_	-	-	550	500
δ_ONO_	-	-	-	378

**Table 2 molecules-28-04676-t002:** The most characteristic IR vibrations (cm^−1^) of salinomycinic acid and complexes **1b**/**2b** in KBr pellets or CHCl_3_ solution (the theoretically predicted scaled values are given in parentheses).

Vibration	SalH	1b	2b
	KBr/CHCl_3_	KBr/CHCl_3_	KBr/CHCl_3_
$\nu_{OH}$_,_ OH	3506/3497	3456	3452
ν_C=O_	1710	1710	1710 (1687)
$\nu_{C=O}$, COOH	1710	-	-
$\nu_{C=O}^{asym}$, COO^−^	-	1552	1512 (1484)
$\nu_{C=O}^{sym}$, COO^−^	-	1411	1411 (1423)
ν_C=C_	1638	1632	1637
$\nu_{NO}$, NO_3_^−^	-	-	1552 (1579, 1551)
$\nu_{NO}$, NO_3_^−^	-	-	1288 (1253)
ν_MO_	-	530	540
δ_ONO_	-	-	380

**Table 3 molecules-28-04676-t003:** Metal–oxygen bond lengths (in Å) for **2a**–**b**.

M-O Bond	Populated Conformers of Complex 2a		M-O Bond	Populated Conformers of Complex 2b	
	39.3%	25.9%		47.7%	32.7%
O1-Ce	2.42	2.40	O1-Ce	2.40	2.40
O2-Ce	2.54	2.45	O2-Ce	2.42	2.40
O8-Ce	2.55	2.68	O4-Ce	2.67	2.64
O9-Ce	2.61	2.76	O5-Ce	2.58	2.55
O11-Ce	2.46	2.45	OH^−^-Ce	2.04	2.08
OH^−^-Ce	2.05	2.07	η_2-_NO_3_^−^-Ce	2.44 2.50	2.49 2.49
NO_3_^−^-Ce	2.48	2.45			
η_2-_NO_3_^−^-Ce	2.40 2.52	2.45 2.48	η_2-_NO_3_^−^-Ce	2.46 2.55	2.46 2.53

**Table 4 molecules-28-04676-t004:** Minimum inhibitory concentration of MonH × H_2_O, SalH and complexes **1**–**2**.

Compound	*K. rhizophila* ATCC 9341		*B. subtilis* ATCC 6633		*B. cereus* ATCC 11778	
	µg/mL	µM	µg/mL	µM	µg/mL	µM
MonH × H_2_O	125	181.4	15.63	22.7	3.91	5.7
**1a**, [Ce(Mon)_2_(OH)_2_]	500	326.4	15.63	10.2	7.81	5.1
**2a**, [Ce(Mon)(η_2_-NO_3_)(NO_3_)(OH)]	500	525.8	62.5	65.7	15.63	16.4
SalH	62.5	83.2	31.25	41.6	7.81	10.4
**1b**, [Ce(Sal)_2_(OH)_2_]	62.5	37.3	31.25	18.7	7.81	4.7
**2b**, [Ce(Sal)(η_2_-NO_3_)_2_(OH)]	62.5	60.6	31.25	30.3	7.81	7.6
(NH_4_)_2_[Ce(NO_3_)_6_]	>1000	>1824	>1000	>1824	>1000	>1824

**Table 5 molecules-28-04676-t005:** CC_50_ values and selectivity index (SI) of target compounds and clinically approved anticancer therapeutics for Hela and Lep-3 cell lines (MTT test, 72 h treatment).

Compound	MM, g/mol	HeLa		Lep-3		SI
		CC_50_, µg/mL	CC_50_, µM	CC_50_, µg/mL	CC_50_, µM	
MonH × H_2_O	688.87	0.52	0.75	3.56	5.17	6.89
**2a**, [Ce(Mon)(η_2_-NO_3_)(NO_3_)OH]	951.00	<0.50	<0.53	6.00	6.31	>12.62
SalH	751.01	1.43	1.90	>10	>13.32	>7.02
**2b**, [Ce(Sal)(η_2_-NO_3_)_2_OH]	1031.13	2.81	2.73	>10	>9.71	>3.56
cisplatin [62]	300.05	8.46	28	0.49	1.63	0.058
oxaliplatin [62]	397.29	6.12	15	0.95	2.39	0.159
epirubicin [62]	543.53	17.38	32	0.74	1.36	0.043

## Data Availability

Data are available from the authors upon request.

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
