# Peer review of "Novel Cerium(IV) Coordination Compounds of Monensin and Salinomycin"

_molecules, 2023, doi:10.3390/molecules28124676_

Round 1
Reviewer 1 Report
The paper reports on the synthesis of two novel coordination compounds of cerium with monensin amd salinomycin and the study of their properties and biological activity. Good antebactirial properties and selective cytotoxicity of these compounds against cancer cells make them promisinig candidates for development of future anti-cancer therapies. The subject of the paper fits well the scope of the Molecules journal.
I have the following comments:
1. In the Introduction section of the manuscript, recent advances in the chemistry of cerium(IV) compounds should be discussed.
2. Available data on the cytotoxicity of cerium(IV) species against cancer cells should also be discussed.
3. There are no direct experimental data confirming the oxidation state of cerium in these new compounds. Actually, the syhthetic route proposed could result in a partial or even complete reduction of cerium(IV) species and formation of cerium(III) species. I would suggest XPS analysis of these compounds to resolve this issue. This is of primary importance since Ce(IV) species are mentioned in the title of the manuscript, in the abstract, and in the conclusions.
Minor editing of English language is required
Author Response
Please seet the attachment.
The authors highlighted in blue the all main corrections for easier review purposes.

Reviewer 2 Report
The manuscript presented by Petkov et al. describes the synthesis, characterization and biological evaluation of novel Ce(IV) complexes based on monensin and salinomycin antibiotics. The research work was well conceived, and results are well discussed. However, there are some issues to be addressed:
i) Simplify the Abstract by removing the first two sentences (lines 15-21).
ii) Change “properties” by “characterization” in headings of Sections 2.1 and 2.2.
iii) Present IR and UV-Vis spectra as Supplementary Information.
iv) Remove the NMR characterization data and discussion because no relevant information can be obtained from the poor resolved spectra presented. Did the authors try to perform the spectra in a different solvent (e.g. DMSO).
v) Present MIC values in mg/mL rather than in µM.
vi) Change “Table 3” by “Table 4” in line 401.
vii) Why choosing non-pathogenic K. rhizophila and B. subtilis bacteria? Why were no Gram-negative bacteria tested?
viii) Present Figure 12 as Supplementary Information.
ix) Provide a clear explanation for the highly deviation observed between calculated elemental analyses values and those obtained experimentally. If not possible, provided the mass spectra of the compounds.
x) Provide the missing C and H elemental analyses values for complex 1b.
xi) Change “….excess of antibiotics” by “…1:2 reaction of (NH4)2[Ce(NO3)6] with the antibiotics…” in line 688.
Author Response
Please see the attachment.
The authors highlighted in blue the all main corrections for easier review purposes.

Round 2
Reviewer 1 Report
The authors addressed most of my comments. The paper is now suitable for publication in Molecules.
Minor editing of English language is required.
Reviewer 2 Report
The manuscript was reviewed taking in consideration all suggestions presented to the authors. Thus, I would accept it in the present form.